# Decision-Focused Evaluation of Worst-Case Distribution Shift

**Kevin Ren**[1]                    **Yewon Byun**[2]                    **Bryan Wilder**[2]

[1]Statistics and Data Science Dept., Carnegie Mellon University, Pittsburgh, Pennsylvania, USA
[2]Machine Learning Dept., Carnegie Mellon University, Pittsburgh, Pennsylvania, USA

## Abstract

Recent studies have shown that performance on downstream optimization tasks often diverges from standard accuracy-based losses, highlighting that the loss function of a predictive model should align with the decision task of the downstream optimizer [Wilder et al., 2019, Elmachtoub and Grigas, 2022]. Despite this observation, no work—to our knowledge—has yet examined the impact of this divergence for distribution shift. In this paper, we demonstrate that worst-case distribution shifts identified by traditional average accuracy-based metrics fundamentally differ from those for the downstream decision task at hand. We introduce a novel framework that employs a hierarchical model structure to identify worst-case distribution shifts in predictive resource allocation settings. This task is more difficult than in standard distribution shift settings because of combinatorial interactions, where decisions depend on the *joint* presence of individuals in the allocation task. We show that the problem can be reformulated as a submodular optimization problem, enabling efficient approximations, to capture shifts both within and across instances of the optimization problem. We apply our solution to real-world datasets in public service settings, providing empirical evidence that worst-case shifts for one metric often significantly diverge from worst-case distributions for other metrics.

## 1 INTRODUCTION

In many real-world prediction settings, machine learning algorithms frequently encounter performance degradation due to distribution shifts, which are characterized by statistical differences between the data encountered during deployment and the data used in training [Quinonero-Candela et al., 2008, Zech et al., 2018, Koh et al., 2021]. In particular, we are motivated by resource allocation settings, in which predictions are used to prioritize individuals within a given decision problem to receive a scarce resource. Here, performance drops in unseen populations can lead to both inefficient and inequitable allocation policies, whether they be potentially live-saving treatments for a disease or public service programs to mitigate risks such as unemployment [Singh et al., 2022, Roland et al., 2022].

Developing methods for estimating worst-case distribution shifts [Subbaswamy et al., 2021, Li et al., 2021, Thams et al., 2022, Huang et al., 2022] is crucial to help practitioners identify problems ahead of time, and either mitigate them or re-evaluate whether to proceed with deployment. Similarly, previous works in the distributionally robust optimization (DRO) literature seek to minimize worst-case loss over a feasible set of distributions [Duchi and Namkoong, 2021]. However, these methods universally focus on identifying shifts under which the model suffers a loss in average accuracy, as measured by traditional loss functions. The key motivation for our work is that the worst-case shifts identified by such individual-level processes may *not* coincide with the worst-case shifts for decisions that require decision-focused optimization over an entire group of individuals, which introduces specific objectives and constraints: a model may be more robust than expected if errors do not flip the optimal decision, and conversely less robust if decisions are sensitive to small errors.

As a simple example, consider a decision maker (e.g. the operator of an emergency room) deciding whether or not to administer a limited treatment to a population consisting of two types of individuals (people seeking treatment). For the first type, their outcomes are very noisy; however, they virtually never require intensive treatment. Thus, all decisions are equally good. For the second type, we can predict outcomes with moderate accuracy, but there is substantial variation in treatment needs. A traditional DRO-style algorithm, identifying worst-case shift with respect to an individual-level

*Accepted for the 40th Conference on Uncertainty in Artificial Intelligence* (UAI 2024).

loss, will typically concentrate more probability on individuals of the first type because predictive accuracy is worse for them – this algorithm would solely seek to maximize worst-case alpha-tail performance in the dataset. However, from a resource allocation perspective, a population composed largely of the first type does not impose difficult trade-offs. This is due to the fact that demand for the resource is low; thus, even uninformed predictions will lead to near-optimal decisions. Furthermore, a population composed of the second type of individuals may be much more challenging, even if the model is more accurate per-individual, since finer distinctions must be made when weighing treatment costs and predicted benefits. This matches the intuition behind recent interest in predict-then-optimize settings, where many studies have shown that the loss function of a predictive model ought to be tuned to reflect the decision task undertaken by the downstream optimizer [Wilder et al., 2019, Elmachtoub and Grigas, 2022, Vanderschueren et al., 2022, Mandi et al., 2020]. However, despite the understanding that performance on downstream optimization tasks often diverges from standard accuracy-based losses, no work – to our best knowledge – has examined the consequences of this divergence for distribution shift.

Capturing the population-level dependencies of resource allocation tasks necessitates a new approach to modeling potential distribution shifts. Standard approaches model individuals as iid and consider perturbations of the marginal probability associated with each individual. However, the arrivals of different individuals are often not plausibly independent. For example, consider an emergency department attempting to triage patients. Due to factors like seasonality in the frequency of many medical conditions, the arrival of different kinds of individuals are in fact correlated, e.g., seeing one patient with respiratory illness makes it more likely that many other patients with respiratory illness will arrive that day. Moreover, triage decisions are made jointly over the entire set of individuals for each day, not marginally for each individual. Formally, decisions are made on the level of optimization instances that consist of many individuals. Since decisions depend on the joint set of individuals present, we must be able to model shifts in the entire joint distribution, not just the marginal probability of each individual.

To address this, we use a hierarchical model for the data generating distribution to capture the optimization instance-style of modeling real-world allocation tasks. More precisely, we model the task of estimating worst-case allocation outcomes as an optimization problem over a hierarchical model, where shift can take place between optimization instances, as well as between the individuals belonging to each optimization instance. This task is significantly more difficult than in standard distribution shift settings because of combinatorial interactions, where decisions depend on the *joint* presence of individuals in the allocation task. We

show that, by reformulating the aforementioned optimization problem as a submodular maximization task, we are also able to address the complexity of combinatorial interactions.

To summarize, our contributions are as follows:

1. We introduce a novel framework that employs a hierarchical model structure to identify worst-case distribution shifts in predictive allocation settings. We show that the problem can be reformulated as a submodular optimization problem, enabling efficient approximations. This captures shifts both within and across instances of the optimization problem and addresses the complexity of combinatorial interactions.

2. In real world predictive allocation settings (e.g., public service data), we empirically show that worst-case shifts substantially differ from those estimated by traditional methods (i.e., metrics that focus on individual-level accuracy).

Our findings highlight that in order to *safely* build and assess ML systems for high-stakes allocation settings, systems must account for the decision task at hand to avoid overestimating their robustness to distribution shift.

## 2 PROBLEM SETUP

**Predictive Modeling with Downstream Allocation.** We address scenarios in which a decision maker must allocate a limited resource within cohorts of individuals (e.g., an emergency department that must triage individuals who arrive every day). We model this process as a distribution over *instances* of the allocation problem, where each instance is composed of individuals $i$ with their own features $x_i \in \mathbb{R}^{d_1}$ and outcome $y_i \in \mathbb{R}^{d_2}$. Let $X$ be the feature matrix that collects the feature vectors of each individual in a given instance and $Y$ the corresponding outcome matrix. Let $\mathbb{P}$ be the distribution over instances, i.e., $X, Y \sim \mathbb{P}$. We will also sometimes need to refer to the marginal distribution that $\mathbb{P}$ induces over individuals, denoted as $\mathbb{P}_{\text{ind}}$. The decision maker observes training data $\{X_j, Y_j\}_{j=1}^k$, where instance $j$ contains $n_j$ individuals. They select a predictive model $m$ which outputs a prediction $m(X)$ based on the features of each instance. Typical architectures accomplish this by making a separate prediction $m(x_i)$ for each $x_i \in X$ and aggregating the results, but we do not assume this.

The decision maker uses the predictions made by $m$ to solve an optimization problem that models the constrained resource allocation, resulting in an allocation vector $Z$. The goal of this problem is to maximize an objective function $f$ which depends both on the decision $Z$ and on the (unknown) labels $Y$ over a feasible set $\Phi$. Given the predictions $\hat{Y} =$

$m(X)$, the corresponding decision is

$$Z^*(\hat{Y}) = \arg \max_{Z \in \Phi} f(Z, \hat{Y}).$$

We define the *decision loss* on a given instance to be the regret relative to if the true $Y$ were known:

$$DL(Y, \hat{Y}) = g(f(Z^*(Y), Y), f(Z^*(\hat{Y}); Y)).$$

Common choices for $g$ may include subtraction (regret) or division (relative regret).

The goal for our model is to do well in expectation over $\mathbb{P}$, minimizing $\mathbb{E}_{X, Y \sim \mathbb{P}}[DL(Y, m(X))]$.

**Identifying worst-case distribution shifts.** We consider the common setting that the distribution $\mathbb{P}$ over instances of the allocation problem may differ in deployment, compared to what the model has encountered during training. Specifically, we consider the challenge of identifying the worst-case distribution shift for allocation performance within a constrained set parameterized by the total size of the shift allowed. Let $\Theta$ denote such a set of potential distributions. Our objective is to find

$$\arg\max_{P \in \Theta} \mathbb{E}_{X, Y \sim P}[DL(Y, m(X))].$$

Identification of such worst-case distributions is a common objective in order to allow model practitioners to understand and ameliorate potential failures in deployment [Pfohl et al., 2022, Subbaswamy et al., 2021, Li et al., 2021, Thams et al., 2022, Huang et al., 2022]. Previous work considers this problem for standard loss functions, which are *separable* across individuals, i.e., which can be written in the form $\mathbb{E}_{x_i, y_i \sim \mathbb{P}_{\text{ind}}}[\ell(m(x_i), y_i)]$ for some individual-level loss $\ell$ with an expectation taken over the marginal distribution over individuals $\mathbb{P}_{\text{ind}}$. For instance, $\ell$ might be the mean squared error or cross-entropy loss. The key motivation for our work is that the worst-case shifts identified by an individual-level process may *not* coincide with the worst-case shifts for the instance-level decisions.

Modeling and solving the worst shift identification problem for predictive resource allocation requires us to address two new challenges that are not present for previous work at the individual level. First, we must provide a parameterized family of distribution shifts over instances, which are composed of *sets* of individuals. Second, we must develop algorithms to solve the resulting optimization problem over distributions, a task which turns out to be considerably more challenging because of the associated combinatorial structure where the impact of one individual on the loss depends on the presence of other individuals in the set.

**Additional Related Work.** There is a large body of literature that develops real-world resource allocation models, as well as criticisms of their shortcomings when exposed to distributions shift [Verma et al., 2023, Athey et al., 2023,

Wang et al., 2022, Schultz et al., 2019]. Our work can be seen as offering a precise way to operationalize and test for distribution shift concerns before deployment in such settings. Also related is work in statistics on learning robust individual treatment rules. E.g., Mo et al. [2021] devise a set of methods to obtain distributionally robust treatment allocation policies given covariate shift. Our work differs in considering the consequences of joint optimization over a population of individuals, and is the first to account for downstream optimization in assessing worst-case distribution shift..

There is an extensive literature devoted to distribution shift in typical ML settings, as opposed to the resource allocation problems that are our focus. Our work is closest to the challenge of diagnosing worst case shifts [Li et al., 2021, Subbaswamy et al., 2021, Thams et al., 2022]. There is also a great deal of work devoted to training models that are robust to distributions shift via methods like Distributionally Robust Optimization (DRO) [Duchi and Namkoong, 2021, Rahimian and Mehrotra, 2019, Levy et al., 2020], and we build on some techniques from this literature. See Appendix A for further related work.

## 3 METHODS

**Defining a constrained set of shifts.** The first challenge to identifying worst-case distribution shifts for predictive resource allocation is to formulate a model for the set of possible distributions $\Theta$. This is more difficult than in the standard supervised learning setting, where typical approaches define a set centered on the empirical distribution over individuals. For example, common choices include a $f$-divergence or Wasserstein uncertainty set [Namkoong and Duchi, 2016, Kuhn et al., 2019]. Implicitly, such formulations are based on the assumption that individuals are sampled independently from $\mathbb{P}_{\text{ind}}$ and so can be represented just by a vector containing the marginal probability of each individual. However, in our setting the instance-level structure means that individuals are not marginally independent: the patients who all arrive at a hospital on a specific day (forming an instance of the allocation problem) may differ systematically from those who arrive a month later. To account for this, we represent our setting via a two-level generative model. First, a latent instance-level parameter $\xi$ is sampled. We denote the marginal distribution over $\xi$ as $\mathbb{P}_\xi$. Second, individuals within the instance are sampled iid conditional on $\xi$:

$$\xi \sim \mathbb{P}_\xi$$
$$x_i, y_i \overset{\text{iid}}{\sim} \mathbb{P}_{\text{ind}}(\cdot|\xi), i = 1 \dots n_j$$

This represents the assumption that individuals are conditionally independent given instance-level information. For example, after conditioning on circumstances in the community (e.g. current disease levels), we suppose that the individual patients arriving at the hospital are independent.

We remark that some appropriate $\xi$ is guaranteed to exist by De Finetti's theorem so long as individuals are modeled as exchangeable [Orbanz and Teh, 2010].

Given this generative model, we adapt the common strategy of using the empirical distribution over the samples $\hat{\mathbb{P}}$ as a proxy for the unobserved $\mathbb{P}$. Specifically, we allow both deviation from the empirical distribution over instances (to model shift in $\mathbb{P}_\xi$) as well as deviation from the empirical distribution over individuals within each instance (to model shift in each $\mathbb{P}_{\text{ind}}(\cdot|\xi)$). Let $\xi_j$ be the value of the latent variable in observed instance $j$. Importantly, even though $\xi_j$ is not itself observed, we only need to be able to model shifts in the distribution over $X, Y$ conditional on $\xi_j$. For this purpose, let $\hat{\mathbb{P}}_{\text{ind},j}$ denote the empirical distribution over individuals within observed instance $j$; this will be our empirical proxy for $\mathbb{P}_{\text{ind}}(\cdot|\xi_j)$. If the decision maker happens to have additional samples believed to be from the same population, these could be used as well. Accordingly, we define the feasible set of shifts for $\mathbb{P}_{\text{ind}}(\cdot|\xi_j)$ to be

$$\Theta_j = \{Q_j \,|\, D(Q_j, \hat{\mathbb{P}}_{\text{ind},j}) \leq \rho_{\text{ind}}\}$$

where $D$ is a standard divergence on distributions (e.g., the $\chi^2$ divergence) and $\rho_{\text{ind}}$ is a parameter chosen by the user to control the amount of distribution shift allowed at the individual level. To obtain the overall set of feasible shifts, we additionally allow a controlled level of shift in the distribution over instances. Let $\hat{\mathbb{P}}_\xi$ be the empirical distribution over the sampled instances (emphasizing again that the values of $\xi$ are irrelevant and we treat $\hat{\mathbb{P}}_\xi$ just as a distribution over the indices $1...k$). We will represent our feasible set by the combination of a distribution $Q_\xi$ over the sampled instances alongside a set $\{Q_j\}_{j=1}^k$ of within-instance distributions over individuals. The final feasible set is thus

$$\Theta = \{(Q_\xi, \{Q_j\}_{j=1}^k) | D(Q_\xi, \hat{\mathbb{P}}_\xi) \leq \rho_\xi, Q_j \in \Theta_j \; \forall j\}$$

where $\rho_\xi$ is a final parameter specifying the allowed degree of shift across instances. Each element of $\Theta$ defines a distribution, which can be sampled from by first sampling an instance identifier $j$ from $Q_\xi$ and then sampling individuals iid from $Q_j$. $Q_\xi$ can be represented as a vector in $\mathbb{R}^k$ giving the probability of each instance, and $Q_j$ can be represented as a vector of size $n_j$ giving the marginal probability placed on each observed individual.

**Optimization over the set of shifts.** The model above induces an optimization problem to identify the worst-case shift with respect to the decision loss. Specifically, we wish to solve

$$\max_{Q \in \Theta} \mathbb{E}_{j \sim Q_\xi} \left[ \mathbb{E}_{X,Y \sim Q_j} [DL(Y, m(X))] \right]. \quad (1)$$

To analyze the structure further, we expand the expectations into sums, using that samples are iid within instances. Let $S_j$ denote the set of all possible draws (with replacement)

of $n_j$ individuals from the observed samples. The problem becomes

$$\max_{Q \in \Theta} \sum_{j=1}^k Q_\xi(j) \sum_{X,Y \in S_j} \left( \prod_{i=1}^{n_j} Q_j(x_i, y_i) \right) DL(Y, m(X)).$$

A first step towards solving this problem is to note that each $Q_j$ can in fact be computed separately, since the outer objective is a sum with non-negative coefficients. That is, we can define

$$Q_j^* = \arg \max_{Q_j \in \Theta_j} \sum_{X,Y \in S_j} \left( \prod_{i=1}^{n_j} Q_j(x_i, y_i) \right) DL(Y, m(X))$$

and then solve

$$\max_{Q \in \Theta} \sum_{j=1}^k Q_\xi(j) \sum_{X,Y \in S_j} \left( \prod_{i=1}^{n_j} Q_j^*(x_i, y_i) \right) DL(Y, m(X))$$

to obtain the optimal distribution over instances. The outer problem has a relatively tractable structure which is closely related to existing work on distributionally robust optimization. Given knowledge of $Q_j^*$, it can be solved using off-the-shelf convex optimization techniques. However, the inner optimization problem for each instance $j$ is much more difficult as it is a *nonconvex* problem. Indeed, polynomial optimization is in general computationally intractable [Karp, 2010, Cook, 2023]. This structure reflects the fundamental change in perspective from individual-level losses to resource allocation: the decision loss encapsulates the joint dependence of decisions on the entire set of individuals who arrive, so the contribution of the parameter for each individual to the loss cannot be neatly disentangled.

To solve this problem, we draw on techniques from the combinatorial optimization literature and prove that it can actually be reformulated as a *DR-submodular* optimization problem. This special structure allows us to develop efficient algorithms with provable approximation guarantees.

**Reformulation as a submodular optimization problem.** We assert that our optimization objective falls under DR-submodular problems – one class of generally non-convex functions. We demonstrate below that under a novel transformation of the objective, our problem is non-monotone DR-submodular in the individual-level and convex in the optimization instance-level. This justifies the use of Frank-Wolfe methods for more optimal solutions to our optimization problems. We assume without loss of generality that the decision loss is either naturally non-positive or bounded in the range $[-\infty, 1]$. Nonnegativity always holds by definition of the regret, and any uniformly bounded loss can be rescaled to an upper value of 1. Thus, for a given loss function $DL$ with such bounds, we define:

$$DL'(Y, \hat{Y}) = DL(Y, \hat{Y}) - 1$$

Then, we introduce a change of variables that supports our non-monotone Frank-Wolfe solution, built on Bian et al.

[2017]. Our solution requires that our feasible set contains the zero vector. Since this condition does not hold for the simplex, we instead optimize over the *offset* from the empirical distribution, where an initialized offset value of 0 is equivalent to the empirical distribution. We define these offsets, along with their feasible sets in the constrained optimization setting, as:

$$W_j(x_i, y_i) = Q_j(x_i, y_i) - \frac{1}{|Q_j|} \; \forall i, j$$

$$\Omega_j = \left\{ W_j \,|\, Q_j + \frac{1}{|Q_j|} \in \Theta_j \right\}$$

This change of variables is finally represented by the following modified optimization problem.

$$\max_{W \in \Omega} \mathbb{E}_{j \sim W_\xi} \left[ \mathbb{E}_{X, Y \sim W_j} [DL'(Y, m(X))] \right] + 1. \quad (2)$$

Note that we apply a correction term of 1 to our final worst-case estimation of $DL'$, in order to account for the adjustment from $DL$ in Equation 1 to $DL'$ in Equation 2.

To begin the justification of this approach, we will prove that an approximate solution to Equation 2 results in an equivalent quality approximation to the original problem in Equation 1. We will then prove that the optimization problem over offsets $W$ is DR-submodular, with full proofs in Appendix B.

**Theorem 3.1.** *Suppose we have a solution $W$ to Equation 2 with value at least $\alpha \cdot OPT'_W - \epsilon$ for some $\alpha \in \mathbb{R}, \epsilon \in \mathbb{R}$, where $OPT'_W$ is the optimal value. $W$ corresponds to a $Q$ with value at least $\alpha \cdot OPT_Q - \epsilon$ where $OPT_Q$ is the optimal value of solving Equation 1.*

In order to prove that the change-of-variables objective is DR-submodular, we first note that the following definition may be helpful:

**Definition 3.2.** *A twice-differentiable function $f : \mathbb{X} \to \mathbb{Y}$ is DR-submodular if all entries of the Hessian matrix are non-positive.*

We then demonstrate that the objective function of our reformulated inner optimization problem satisfies this definition:

**Theorem 3.3.** *The objective function of the optimization problem in Equation 2 is non-monotone DR-submodular in $W_j$.*

Next, we build on this result to introduce efficient approximation algorithms for the reformulated problem.

**Approximation algorithms.** Bian et al. [2017] introduce several methods for maximizing non-monotone DR-submodular problems; we utilize an adaptation of their non-monotone Frank-Wolfe variant in order to solve the above

DR-submodular problems, in the context of resource allocation. Since this algorithm requires that the zero vector be a member of the feasible set, we optimize over the offset from the empirical distribution. Further, we take advantage of strong empirical results identified by past works in applying momentum-based methods to Frank-Wolfe methods [Mokhtari et al., 2018, Li et al., 2020]. These methods store gradients from prior iterations of the algorithm and consider them to adjust the current iteration's gradients.

**Main algorithm.** In our algorithm, we first optimize over individuals within a given optimization instance and then optimize over all optimization instances. We include the full pseudocode in Algorithm 1, a formal writeup of how we implement the Frank-Wolfe algorithm developed by Bian et al. [2017]. Our algorithm includes a subroutine called *gradmax*, which maximizes the dot product over the feasible set of viable allocations and the gradients of the objective function w.r.t. the optimization variables Staib et al. [2019]. Additional implementation details can be found in Appendix C.

---

**Algorithm 1** Frank-Wolfe Method for Maximizing Expected Loss over Optimization Instances

---

**Input:** weight offsets $W_\xi, \cdots, W_j, \cdots, W_k$ (initialized to $\{0\}_1^{n_j}$), $v_0 = \{0\}_1^{n_j}$, iterations, num_samples, num_samples2, $p_t$, $\rho_{\text{ind}}$, $\rho_\xi$
**for** $i = 1$ **to** $k$ **do**
    **for** $j = 1$ **to** iterations **do**
        **for** $r = 1$ **to** num_samples **do**
            sample $\{x_s, y_s\}_{s=1}^{n_j} \sim W_i$
            calculate $\ell = DL'(m(\{x_s\}_{s=1}^{n_j}), \{y_s\}_{s=1}^{n_j})$
            accumulate $\frac{\partial \ell}{\partial W_i}$
        **end for**
        set $v_j = p_t * -\frac{\partial \ell}{\partial W_i} + (1 - p_t) * v_{j-1}$
        solve $\delta = \textbf{gradmax}(v_j, \rho^{\text{ind}}) - \frac{1}{|W_i|}$
        update $W_i = W_i + \frac{1}{\text{iterations}} \delta$
    **end for**
**end for**
initialize $\lambda = \{0\}_1^k$
**for** $i = 1$ **to** k **do**
    **for** $j = 1$ **to** num_samples2 **do**
        sample $\{x_s, y_s\}_{s=1}^{n_j} \sim W_i$
        accumulate $\ell = DL'(m(\{x_s\}_{s=1}^{n_j}), \{y_s\}_{s=1}^{n_j})$
    **end for**
    set $\lambda_i = \text{avg}(\ell)$
**end for**
solve $W_\xi = \textbf{gradmax}(\lambda, \rho_\xi)$

---

## 4 EXPERIMENTS AND RESULTS

We consider the following allocation tasks on real-world data. Specifically, we focus on the following three tasks, utilizing US census data [Ding et al., 2021]. Each task is

motivated by resource allocation problems in a public policy-related setting. More concretely, we consider (1) predicting employment status, (2) predicting an individual's income in American dollars, and (3) predicting whether an individual's income is above or below $50,000 annually. For each task, we consider a hypothetical resource allocation problem, modeling a decision maker who wishes to target a limited intervention to individuals more likely to be unemployed or more likely to have low income, respectively. In each of these tasks, the optimization instances are formed from individuals in a particular US state, reflecting that allocation decisions are made among geographic cohorts and different states may differ systematically in distribution.

We use these domains to run large-scale experiments over thousands of different combinations of models, optimization instances, and loss functions. See Appendix D for further details on the train-test setup, model architectures, etc.

## 4.1 LOSS FUNCTIONS

We identify and compare the worst-case distribution shift for the following loss functions. Recall that the objective of our method is to identify a set of distributions over individuals and optimization instances that maximizes the expected value of a given loss function. First, we look at **top-k**, where the decision maker has $k$ units of the resource available per instance and the objective function is the number of individuals with true label 1 who receive the resource. This is the most canonical model of scarce resource allocation based on predicted risk. Second, we look at **knapsack**, where the decision maker's objective is the same as the top-k setting but they are subject to a knapsack constraint instead of a simple budget [Mulamba et al., 2020, Stuckey et al., 2020, Tang and Khalil, 2022]. More specifically, we set an individual's cost to be proportional to the number of years of education, simulating a policy maker who also wishes to guarantee that public assistance is given preferentially to individuals with less education. Note that top-k is a simplification of knapsack where cost is identical for all individuals. Third, we study a fairness-motivated loss function in which the decision maker takes the decision rendered by top-k but wishes to assess the equity of the resulting allocation. We calculate the true positive rate (TPR) over all distinct racial subgroups in the optimization instance, and calculate a Gini coefficient using these TPRs as a measure of unfairness of treatment over racial groups. For simplicity, we refer to this loss function as **fairness-based loss**.

For the binary prediction tasks, we look at **1-accuracy**, or the misclassification rate over individuals. We also look at the cross entropy loss (**CE**) over individuals. During optimization, CE is scaled as necessary to fit the format required by our Frank-Wolfe solution (i.e. we take the negative inverse of cross-entropy loss).

For the income regression task, we look at mean squared error (**MSE**), which is also scaled as necessary to fit the format required by our Frank-Wolfe sollution (i.e. we take the base-2 log and divide by a constant). Further, we look at a **utility-based loss**, where a decision maker seeks to maximize the Nash social welfare function of income, divided by a constant, over individuals [Kaneko and Nakamura, 1979]. The decision maker also has access to a (finite) budget which prioritizes distributing money to relatively poor individuals to maximize utility.

## 4.2 EXPERIMENTAL SETUP

We train predictive models on each dataset using cross-entropy (CE) loss, mean-squared error (MSE) loss, and Smart Predict-then-Optimize loss (SPO) with knapsack as the underlying decision task [Elmachtoub and Grigas, 2022, Tang and Khalil, 2022]. We train two separate models for each state, one with the SPO loss and one with either CE (for binary classification models) or MSE (for regression models). For each predictive model, we then identify worst-case distributions over all optimization instances, w.r.t. each applicable metric, to obtain different worst-case distributions to compare systematically. Per each of 5 loss functions, this results in 50 worst-case distributions per base model, for 75,000 worst-case distributions over each of the three prediction tasks. After identifying all worst-case distributions, we evaluate each converged worst-case distribution on all *other* applicable metrics. This is accomplished by (i) drawing samples from the generative model for each worst-case distribution, (ii) evaluating the average loss of these samples when inputted to each of the other loss functions, and (iii) for each model, compiling these averages for all worst-case distributions – resulting in the following matrices in Figure 1. See Appendix D for additional experimental details, and Appendix E for additional results.

## 4.3 RESULTS

Given a worst-case distribution w.r.t. one metric, we evaluate the expected value of all other applicable metrics using samples from the distribution. This allows us to quantitatively assess whether a worst-case shift w.r.t. a decision-blind metric (e.g., 1-accuracy, MSE, CE) is also worst-case w.r.t. a decision-based metric (e.g. top-k, knapsack, fairness-based loss, utility-based loss). We refer to decision-based metrics as metrics that consider the relative predictive model outputs and/or individual-level features in an optimization problem. We refer to decision-blind metrics that do *not* consider the relative predictive model outputs; these metrics can be expressed as a sum of losses over individuals. Note that worst-case shifts with respect to the decision-blind metrics can be seen as implementing the standard f-divergence approach popularized by Namkoong and Duchi [2016] [Subbaswamy et al., 2021, Li et al., 2021]. We first present these

results for each allocation task, and then provide a more qualitative interpretation of how worst-case distributions differ by loss function.

In Figure 1, we observe the performance of worst-case distributions w.r.t. each metric (both decision-blind and decision-based) on all other metrics. We find that worst-case distributions w.r.t. a given metric tend to *overfit* on that metric. In other words, for a given dataset and prediction task, the worst-case distribution w.r.t. a metric (e.g. CE) tends to perform comparatively worse on other metrics (e.g. top-k) than the worst-case distribution w.r.t. that other metric. In other words, the expected value of top-k induced by sampling individuals from the worst-case distribution w.r.t. top-k will be higher than the expected value of top-k induced by sampling individuals from the worst-case distribution w.r.t. CE. In particular, this can be observed from the main diagonal entries throughout all subfigures in Figure 1: no worst-case distribution achieved a metric that was (noticeably) higher than the worst-case distribution trained w.r.t. that metric. We find that this trend is generally consistent across all considered metrics, models, and datasets/prediction tasks.

These observations lend great importance to considerations of the downstream allocation tasks predictive models may face when deployed in the real world. For instance, when a practitioner trains a robust model w.r.t one metric (decision-based *or* decision blind), if the downstream allocation problem is changed (even subtly) (e.g. changing the cost per individual from constant to varying based on individual features), in many settings, models could break under shifts that were not initially thought to be worst-case.

We find that the use of decision-blind metrics (MSE, CE, accuracy) to inform judgement on worst-case outcomes for decision-based tasks, in general, risks underestimating true worst-case outcomes, which we can justify with thorough examination of the individuals in our optimization instances. Through these results (see Figures 1, 2, 3), we find that existing methods of identifying worst-case distributions (e.g. DRO), which generally focus on *decision-blind* metrics and optimize worst-alpha-tail performance over individuals, fail to accurately depict worst-case outcomes in the predictive resource allocation setting, wherein model predictions are passed into higher-level optimization problems for decision-making. This implies that our method identifies worst-case distributions for allocative tasks *substantially better than existing DRO-like methods*, as our method more precisely considers the structure of downstream (allocation) problems in our solutions. Below we discuss potential mechanisms for this behavior by analyzing which types of individuals are more likely to be sampled in worst-case distributions w.r.t. a given metric, and how these individuals change given a worst-case distribution w.r.t. a different metric.

For further analysis, we examine differences between the worst-case distributions w.r.t. a selection of metrics by vi-

sualizing the converged weights of individuals of varying feature values and varying model predictions (see Figures 2, 3). We find that, given the same individuals within the same optimization instance, worst-case distributions w.r.t. different metrics differ in the individuals they tend to upweight. We clearly observe that in a worst-case distribution w.r.t CE (Figure 2a), individuals are weighed directly proportionally to their distance from the decision boundary (e.g. false positives with high model predictions receive high weights, as do false negatives with low model predictions). We can use similar logic when identifying which individuals are upweighted in the worst-case distribution w.r.t. MSE (Figure 3a); individuals are gradually upweighted the higher their residual is, or in other words, the farther their prediction strays from their ground-truth label (noted by the dotted identity lines in Figure 3).

In contrast, worst-case distributions w.r.t. decision-based metrics upweight systematically different sets of individuals. We observe that in Figure 2b, two particular types of individuals are highly upweighted: a non-white, positive (i.e. unemployed) individual with a high model prediction, along with a white, negative (i.e. employed) individual with a high model prediction. When these individuals are considered in conjunction within the fairness metric, we tend to correctly treat many positive non-white individuals, incorrectly treat many negative white individuals, and incorrectly fail to treat some positive white individuals. Notice that while non-white individuals tend to be correctly treated here, white individuals are consistently assigned the incorrect treatment, thus achieving high disparity in the quality of treatment between non-white and white individuals and a high fairness-based loss metric. Turning our focus to the utility-based loss metric with income regression in Figure 3b, the worst-case distribution seems to upweight only relatively high-residual individuals, as expected by definition of the utility-based loss metric. More concretely, since our Nash social welfare function scales with the log of income, there exists a diminishing returns effect of allocating money to high-income individuals. Therefore, we can maximize the impact of an ideal treatment on individuals, and thus, achieve higher levels of relative regret, if our sample consists of many low-income individuals, but we instead choose to mistakenly treat higher-income individuals whose predicted income is particularly low. Furthermore, individuals with relatively high values for both ground-truth labels and predictions do not receive high weights despite having large residuals, since they still have higher predicted incomes than the upweighted individuals and are unlikely to be treated in this allocation setting.

Based on the observed differences between the worst-case distributions w.r.t. decision-blind metrics and worst-case distributions w.r.t. decision-based metrics, we conclude that practitioners who use current off-the-shelf DRO methods for building robust predictive allocation models *may not ac-*

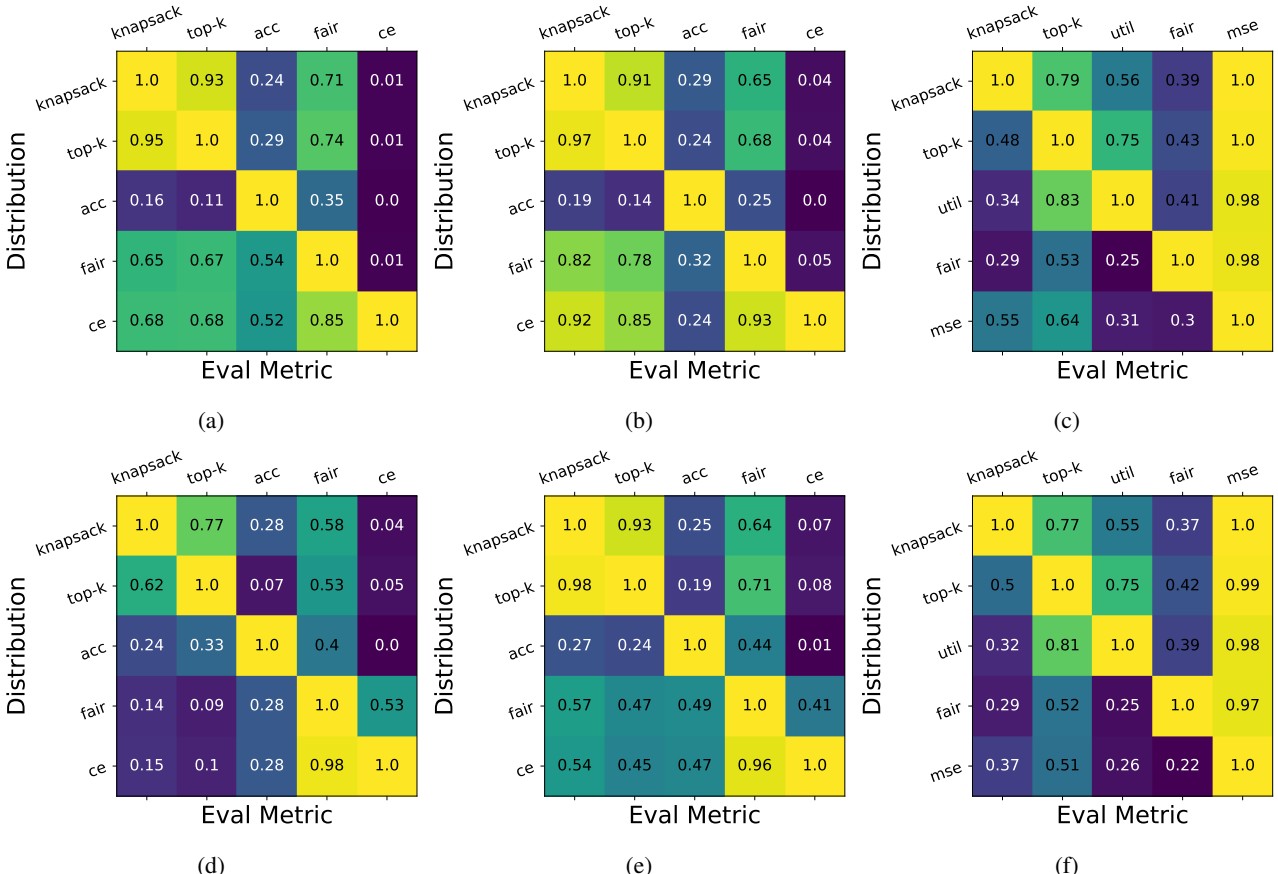

Figure 1: Diagonal-normalized aggregated heat maps over states for models trained with CE loss (in the regression case, MSE loss) (top row) and SPO loss (bottom row). From left to right in each row, results are displayed by task for (a,d) employment classification, (b,e) income classification, and (c,f) income regression. Within each heat map, rows denote the metric the worst-case distribution maximizes, and columns denote the metrics the worst-case distribution was evaluated on. Note that each column is divided by the diagonal entry in that column, resulting in a main diagonal of all 1.0. Furthermore, because CE loss is always negative, each entry in columns corresponding to CE loss is equal to the diagonal entry in that column divided by the original loss in that cell. The strong main diagonals here accentuate our observation that worst-case distributions w.r.t. a given metric tend to 'overfit' on that metric. Note that cross-entropy and accuracy are used only in binary classification tasks, and mean-squared error and utility-based loss are used only in the income regression task.

*tually be training models robust to the true worst-case shifts in the downstream decision task their model will be used for.* In other words, the DRO equivalent of greedily placing weight on poorly-performing individuals w.r.t. decision-blind metrics results in worst-case distributions metrics that fundamentally differ in composition from the worst-case distributions for an optimization-level, decision-based metric. These results provide insights to practitioners deploying robust predictive models for resource allocation: allocative tasks make structurally different uses of machine learning predictions than decision-blind prediction tasks, and methods for identifying distribution shifts must reflect this structure.

Finally, we empirically study the efficiency of our algorithm via empirical comparisons with existing polynomial solvers, in identifying distributions over optimization instances and

individuals. In Figure 4, we evaluate our algorithm for solving Equation 2, as the number of optimization problems sampled during each iteration of the Frank-Wolfe algorithm increases. Note that, by enumerating over all possible sets of $n_j$ individuals and calculating the value of loss for each set, Equation 2 becomes a closed-form polynomial optimization problem. We find that we *consistently* approach values found by the polynomial optimization solver, Pyomo with optimizer IPOPT, when used to solve Equation 2 [Hart et al., 2017, Biegler and Zavala, 2009]. In Figure 4, we plot our estimated solutions to Equation 2 as a proportion of the solutions achieved by Pyomo/IPOPT against the number of samples per iteration of the Frank-Wolfe algorithm. We then aggregate these results by metric across models trained with CE, MSE, or SPO loss functions. This analysis is conducted on all tasks: employment prediction, income

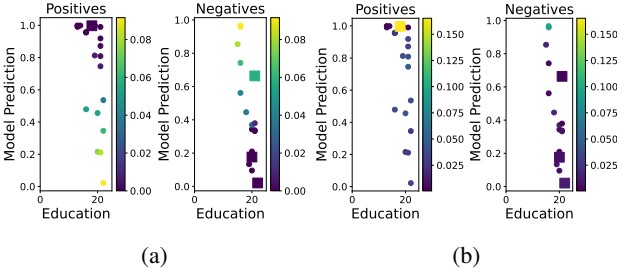

(a)                          (b)

Figure 2: Plots of individuals in an optimization instance in the employment prediction task, from the perspective of worst-case distributions w.r.t. CE (a) and the fairness-based metric (b). The underlying predictive model was trained with CE loss. For each worst-case distribution w.r.t. the metric of interest, we display over all individuals their model predictions, assigned weights, and education level, split by true label. Note that the color-bar denotes the weight in the worst-case distribution and that differently-shaped points represent individuals of different races (circle for white, square for non-white).

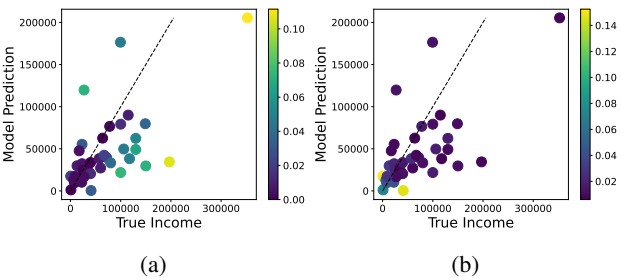

(a)                          (b)

Figure 3: Plots of individuals in an optimization instance in the income regression task, from the perspective of worst-case distributions w.r.t. mse (a) and the utility-based metric (b). The underlying predictive model was trained with mean-squared error loss. For each worst-case distribution, we display over all individuals their model predictions, label income, and assigned weights. In each figure the identity line (True Income = Model Prediction) is marked with a dotted line.

prediction, and income regression. We find that 87% of the plotted curves exceeded 80% of the solutions reached by Pyomo/IPOPT, and all curves reached over 60% of this value. Furthermore, all curves increased monotonically (i.e., closer to the solution reached by Pyomo/IPOPT) as the number of samples increased, indicating that as our method has access to more instances of optimization problems it more effectively maximizes estimates of worst-case loss. The theoretically guaranteed approximation ratio for nonmonotone submodular optimization with Frank-Wolfe algorithms is $\frac{1}{e} \approx 37\%$ [Bian et al., 2017]. Therefore, these results suggest that our algorithm empirically performs much stronger than the worst-case theoretical guarantees.

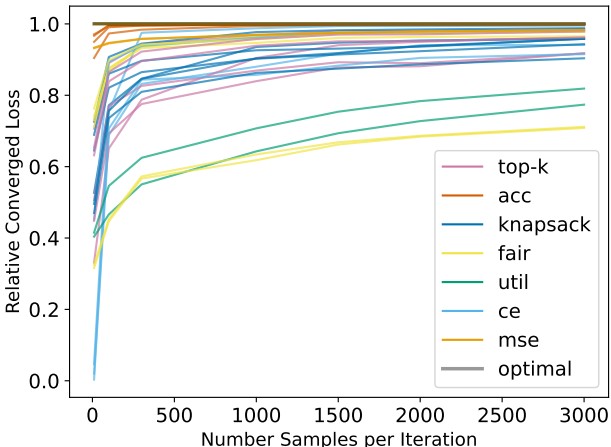

Figure 4: Aggregate results of efficiency experiment. The converged expected values of loss are plotted over all metrics and all datasets. Here $n_j = 8$, and we calculate reference worst-case expected values of each metric using Pyomo with the IPOPT solver. All values are normalized w.r.t. the value of the Pyomo/IPOPT solution. The bolded flat line at $y = 1$ represents the Pyomo/IPOPT solutions. The colored lines each represent, for a given predictive model training method, prediction task, and metric, how closely our method asymptotically reaches the solution quality of the Pyomo/IPOPT solutions. We find that the majority of our curves converged asymptotically to over 80% of the Pyomo/IPOPT solution.

## 5   DISCUSSION

In this paper, we develop an algorithmic approach to find worst-case distribution shifts over a constrained set for predictive resource allocation problems. Our formulation reflects the structure of predict-then-optimize settings, allowing us to account for distribution shift within and between optimization instances through a two-level generative model, which DRO-style methods do not account for. We show that the optimization of this model can be formulated as a submodular optimization problem and solved with a Frank-Wolfe algorithm. We empirically demonstrate that worst-case distributions with respect to decision-blind and decision-based metrics exhibit substantial divergences, and that worst-case distributions with respect to decision-blind metrics may be very far from the worst-case for decision tasks. Finally, we find that our methods efficiently approach solutions found by existing polynomial solvers much more closely than might be suggested by worst-case theoretical guarantees. In all, our results highlight that evaluation of the robustness of ML models in high-stakes resource allocation settings must account for the nature of the downstream decision problem in order to capture the true consequences of potential distribution shift.

# 6 ACKNOWLEDGEMENTS

This work was supported by the AI2050 program at Schmidt Sciences (Grant G-22-64474) and by the AI Research Institutes Program funded by the National Science Foundation under AI Institute for Societal Decision Making (AI-SDM), Award No. 2229881.

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

# Decision-Focused Evaluation of Worst-Case Distribution Shift
## (Supplementary Material)

**Kevin Ren**[1]                **Yewon Byun**[2]                **Bryan Wilder**[2]

[1]Statistics and Data Science Dept., Carnegie Mellon University, Pittsburgh, Pennsylvania, USA
[2]Machine Learning Dept., Carnegie Mellon University, Pittsburgh, Pennsylvania, USA

## A  RELATED WORK

**Methods of Representing Distribution Shift**. Prior studies on distribution shift have investigated some combination of covariate shift, label shift, and/or subpopulation shift in the context of standalone machine learning models Liu et al. [2023]. The concept of subpopulation shift, which relates tangentially to our use of optimization instances to separate individuals, involves a shift in the distribution of data amongst an unseen class variable. This concept has deep ties to fair and robust machine learning, as in practice many models fail to generalize well to specific subgroups within a broader population Yang et al. [2023]. In terms of optimization, beyond the mere diagnosis of distribution shift, via such methods as outlier detection, prior work also encapsulates the development of models that are trained in an adversarial manner such as to minimize their expected loss over a set of 'reasonable' or viable out-of-distribution test sets, known as Distributionally Robust Optimization (DRO) Duchi and Namkoong [2021], Rahimian and Mehrotra [2019]. [Levy et al., 2020] for instance identify a method for DRO with respect to convex loss functions that scales independently of population size, utilizing $\chi^2$ divergence uncertainty sets. In the field of biodiversity Mäkinen and Vanhatalo [2018] identify shifts in the spaciotemporal distribution of arctic species using Bayesian Hierarchical Models tiered on observations, density processes and process parameters, a related paradigm to our use of hierarchical modeling to describe distributions over sets of individuals.

**Predictive Treatment Allocation**. A wide body of work exists to identify optimal treatment allocation policies, particularly in the health and public policy sectors. In terms of applications in public policy, Verma et al. [2023] deploy the first Restless Multi-Armed Bandit framework in public health in India. Their model is used by the NGO ARMMAN to identify women in underserved communities to contact regarding maternal and childcare information. Athey et al. [2023] follow in this more empirical style of evaluation, developing a set of empirically validated treatment strategies to provide 'nudges' to college students in order to increase financial aid application renewals. Finally a smaller body of work exists criticizing the overeagerness of industry in adopting predictive allocation models prior to addressing theoretical and empirical concerns. Wang et al. [2022] cite distribution shift as one major obstacle the field of predictive optimization as a whole must surmount before it is accepted as a legitimate machine learning practice, motivating in particular our work in non-parametric shifts. In the same vein of concerns regarding distribution shift in treatment allocation Schultz et al. [2019] suggest that triage risk models tend to perform better on middle-aged patients than older patients in terms of short-term mortality risk.

**Predict-then-Optimize Machine Learning**. Predict-then-optimize is a more generalized class of problems than resource allocation, in which a model predicts a cost vector from a vector of features, and second, the predicted cost vector is used as a set of parameters for an optimization problem. This class of machine learning along with deterministic optimization is common in many areas of deployment, from limited resource allocation problems to shortest-path problems; in many such setups, we have some unknown cost variable that can be estimated using some machine learning model and optimized through well-known algorithms Elmachtoub and Grigas [2022].

A large portion of work in predict-then-optimize machine learning has sought to train models that perform well in-distribution Wilder et al. [2019], Elmachtoub and Grigas [2022], with significant work aiming to develop loss functions that take into account the optimization task in-context. This framework has also been investigated for specific prediction models along with applications in reinforcement learning and solving combinatorial problems Elmachtoub et al. [2020], Mandi et al. [2020], Wang et al. [2021]. However, a comparatively smaller body of work exists to train or fine-tune machine learning models in

*Accepted for the 40$^{th}$ Conference on Uncertainty in Artificial Intelligence* (UAI 2024).

predict-then-optimize settings that are distributionally robust. Notably Johnson-Yu et al. [2023] examine robust methods for accounting for label shift in predict-then-optimize, proposing modifications for a two-stage machine, decision-focused learning setup to anticipate label shift.

# B  FULL PROOFS OF DR-SUBMODULARITY

**Definition B.1.** *Given some index $i$ and some sample of entries $X, Y$ from a probability distribution $\mathbb{P}$, let $\#(i, (X, Y)) : (\mathbb{Z}, \mathbb{P}) \to \mathbb{N}$ be the number of times the $i$th indexed individual occurs in the sample $X, Y$.*

Before the proof of Theorem 3.1 , we begin by proving the following lemma to establish guarantees on a solution to a modification of Equation 2 that uses the same version of decision loss, $DL$, as Equation 1. This helps us establish bounds on guarantees after changing $DL$ to $DL'$ in Equation 2.

**Lemma B.2.** *Suppose we have a solution $W$ to a slight modification of Equation 2 with value at least $\alpha \cdot OPT_W - \epsilon$ for some $\alpha \in \mathbb{R}, \epsilon \in \mathbb{R}$, where $OPT_W$ is the optimal value, and $DL$ is used in Equation 2 instead of $DL'$. Then, we have a solution to the original optimization problem over $Q_j$, Equation 1, with value at least $\alpha \cdot OPT_Q - \epsilon$ where $OPT_Q$ is the optimal value of the original problem.*

**Proof of Lemma B.2**.

*Proof.* First, formally define the optimization problem representing the modified Equation 2 as

$$\max_{W \in \Omega} \mathbb{E}_{j \sim W_\xi} \left[ \mathbb{E}_{X, Y \sim W_j} \left[ DL(Y, m(X)) \right] \right] + 1.$$

Let the optimal solution to the modified Equation 2 be represented as $OPT_W$. Furthermore, we define the achieved loss of our solution on the modified Equation 2 as $\hat{\phi}$, e.g.

$$\hat{\phi} = \mathbb{E}_{j \sim W_\xi}[\mathbb{E}_{X, Y \sim W_j}[DL(Y, m(X))]]$$

Let $\phi$ denote the realized loss of $W$ on Equation 1 when we transform $W$ into a set of valid probability distributions, e.g.

$$\phi = \mathbb{E}_{j \sim W_\xi + \frac{1}{k}}[\mathbb{E}_{X, Y \sim W_j + \frac{1}{|W_j|}}[DL(Y, m(X))]]$$

Using these definitions,

$$
\begin{aligned}
\hat{\phi} &= \sum_{j \in W_\xi} (W_\xi(j) + \frac{1}{k}) \sum_{X, Y \in S_j} DL(Y, m(X)) Pr(X, Y) \\
&= \sum_{j \in W_\xi} (W_\xi(j) + \frac{1}{k}) \sum_{X, Y \in S_j} DL(Y, m(X)) \prod_{i \in (X, Y)}^{n_j} (W_j(X_i, Y_i) + \frac{1}{|W_j|}) \\
&= \sum_{j \in W_\xi + \frac{1}{k}} (W_\xi + \frac{1}{k}) \sum_{X, Y \in S_j} DL(Y, m(X)) \prod_{i \in (X, Y)}^{n_j} (W_j(X_i, Y_i) + \frac{1}{|W_j|}) \\
&= \mathbb{E}_{j \sim W + \frac{1}{k}}[\mathbb{E}_{X, Y \sim W_j + \frac{1}{|W_j|}} DL(Y, m(X))] \\
&= \phi.
\end{aligned}
$$

With the same logic, we substitute aspects of the sampling process using $W$ to derive information about the optimal solutions:

$$OPT_W = \max_{W \in \Omega} \mathbb{E}_{j \sim W_\xi}[\mathbb{E}_{X,Y \sim W_j}[DL(Y, m(X))]]$$

$$= \max_{W \in \Omega} \sum_{j \in W_\xi} (\frac{1}{k} + W_\xi(j)) \sum_{X,Y \in S_j} Pr(X,Y)DL(Y,m(X))$$

$$= \max_{W \in \Omega} \sum_{j \in W_\xi} (\frac{1}{k} + W_\xi(j)) \sum_{X,Y \in S_j} DL(Y,m(X)) \prod_{i \in (X,Y)}^{n_j} (\frac{1}{|W_j|} + W_j(X_i, Y_i))$$

$$= \max_{Q \in \Theta} \sum_{j \in Q_\xi} Q_\xi(j) \sum_{X,Y \in S_j} DL(Y,m(X)) \prod_{i \in (X,Y)}^{n_j} Q_j(X_i, Y_i)$$

$$= OPT_Q.$$

Thus,

$$\hat{\phi} \geq \alpha(OPT_W) - \epsilon \implies \phi \geq \alpha(OPT_Q) - \epsilon.$$

$\square$

**Proof of Theorem 3.1**.

*Proof.* Let the final expectation of our estimated solution to Equation 2 be $\hat{\phi}'$, e.g.

$$\hat{\phi}' = \mathbb{E}_{j \sim W_\xi}[\mathbb{E}_{X,Y \sim W_j}[DL'(Y, m(X))]]$$

Furthermore, let $OPT'_W$ denote the optimal solution to Equation 2. We have:

$$\hat{\phi}' \geq \alpha(OPT'_W) - \epsilon$$

$$= \alpha(\max_{W \in \Omega} \left( \sum_{j \in W_\xi} W_\xi(j) \sum_{X,Y \in S_j} Pr(X,Y)(DL'(Y,m(X)) - 1) \right) + 1) - \epsilon$$

$$= \alpha(\max_{W \in \Omega} \left( \sum_{j \in W_\xi} W_\xi(j) \left[ \sum_{X,Y \in S_j} Pr(X,Y)DL'(Y,m(X))) - \sum_{X,Y \in S_j} Pr(X,Y) \right] \right) + 1) - \epsilon$$

$$= \alpha(\max_{W \in \Omega} \left( \sum_{j \in W_\xi} W_\xi(j)(-1 + \sum_{X,Y \in S_j} Pr(X,Y)DL'(Y,m(X))) \right) + 1) - \epsilon$$

$$= \alpha(\max_{W \in \Omega} \left( 1 + \sum_{j \in W_\xi} -W_\xi(j) + \left[ \sum_{j \in W_\xi} W_\xi(j) \sum_{X,Y \in S_j} Pr(X,Y)DL'(Y,m(X)) \right] \right))$$

$$= \alpha(\max_{W \in \Omega} \left( \sum_{j \in W_\xi} W_\xi(j) \sum_{X,Y \in S_j} Pr(X,Y)DL'(Y,m(X)) \right))$$

$$= \alpha(OPT'_W) - \epsilon \implies$$

$$\hat{\phi}' \geq \alpha(OPT_Q) - \epsilon \qquad \text{(by Lemma B.2)}$$

$\square$

**Proof of Theorem 3.3**.

*Proof.* Consider the gradient of the expansion of Equation 2 into sums w.r.t. a single weight placed on an individual of index $a$ in instance $j$:

$$\frac{\partial}{\partial W_j(x_a, y_a)} \sum_{X,Y \in S_j} \left[ \prod_{i=1,}^{n_j} \left( W_j(x_i, y_i) + \frac{1}{|W_j|} \right) \right] DL'(Y, m(X))$$

$$= \sum_{X,Y \in S_j} \#(a, (X,Y))(W_j(x_a, y_a) + \frac{1}{|W_j|})^{\#(a,(X,Y))-1} \left[ \prod_{i=1,i\neq a}^{n_j} \left( W_j(x_i, y_i) + \frac{1}{|W_j|} \right) \right] DL'(Y, m(X))$$

$$= \mathbb{E}_{X,Y \sim W_j}[DL'(Y, m(X)) \frac{\#(a, (X,Y))}{W_j(x_a, y_a) + \frac{1}{|W_j|}}]$$

The resulting expectation is non-positive due to non-negative weights in any probability mass function and non-positive loss $DL$. Next consider the diagonal entries of the Hessian matrix of the refactored objective:

$$\frac{\partial^2}{\partial^2 W_j(x_a, y_a)} \sum_{X,Y \in S_j} \left[ \prod_{i=1}^{n_j} \left( W_j(x_i, y_i) + \frac{1}{|W_j|} \right) \right] DL'(Y, m(X))$$

$$= \sum_{X,Y \in S_j} \#(a, (X,Y))(\#(a, (X,Y)) - 1)(W_j(x_a, y_a) + \frac{1}{|W_j|})^{\#(a,(X,Y))-2} \left[ \prod_{i=1,i\neq a}^{n_j} \left( W_j(x_i, y_i) + \frac{1}{|W_j|} \right) \right] DL'(Y, m(X))$$

$$\leq 0. \qquad \qquad \text{(by non-positive DL')}$$

Finally consider the off-diagonal entries of the Hessian matrix below, where we consider a second arbitrary individual of index $b$, where $a \neq b$:

$$\frac{\partial^2}{\partial W_j(x_a, y_a)\partial W_j(x_b, y_b)} \sum_{X,Y \in S_j} \left[ \prod_{i=1}^{n_j} \left( W_j(x_i, y_i) + \frac{1}{|W_j|} \right) \right] DL'(Y, m(X))$$

$$= \sum_{X,Y \in S_j} \#(a, (X,Y))(\#(b, (X,Y)))(W_j(x_a, y_a) + \frac{1}{|W_j|})^{\#(a,(X,Y))-1}$$

$$(W_j(x_b, y_b) + \frac{1}{|W_j|})^{\#(b,(X,Y))-1} \left[ \prod_{i=1,i\notin\{a,b\}}^{n_j} \left( W_j(x_i, y_i) + \frac{1}{|W_j|} \right) \right] DL'(Y, m(X))$$

$$\leq 0. \qquad \qquad \text{(by non-positive DL')}$$

By Definition 4.3 $J$ is DR-submodular and by Definition 4.2 $J$ is non-monotone; this conclusion applies WLOG to all subpopulations between 1 and $k$ inclusive. $\square$

# C   ADDITIONAL METHODOLOGICAL DETAILS

- Due to the strong empirical improvements we saw along with past work by such works as Li et al. [2020], we also introduce a momentum term into the update rule that preserves a portion of gradients calculated in the previous iteration of the algorithm, initialized to 0.

- Building on prior work from Staib et al. [2019], Frank-Wolfe algorithms commonly require subroutines to maximize the dot product over the feasible set of viable allocations and the gradients of the objective function with respect to the optimization variables. We incorporate their work as a subroutine within our algorithm, termed *gradmax*. Gradmax is also used to solve the optimization problem over all optimization instances, where we input a vector of converged worst-case losses for all optimization instances along with $\rho_\xi$ into gradmax, which then returns the probability distribution within the feasible set that maximizes expected worst-case loss over optimization instances.

# D   ADDITIONAL EXPERIMENTAL DETAILS

Regarding the training process of our underlying predictive models:

- For each of the 50 US states in the census datasets we train two models: one with CE loss (for regression, MSE) and one with SPO loss, resulting in 300 total models.

- Note that, while training with cross-entropy/mean-squared error loss can be accomplished with nothing but the raw train set, training with SPO loss requires treating each optimization instance as a random sample of individuals from the train set. To this end, we take 15,000 samples of $n_j = 40$ individuals from the combined training sets of each state, each sample representing one resource allocation problem.

- Two such models are trained on each of the three datasets, resulting in six total trained models per each of the fifty optimization instances (300 total predictive models).

- The predictive models used to predict employment/income also differ in architecture depending on the dataset. For employment classification we train logistic regression models on each state's train set; for income classification/regression we train a 2-hidden-layer neural network. All categorical features are mapped to Pytorch embedding layers that are also trained for each predictive model.

Regarding the application of these predictive models in finding worst-case distributions w.r.t. relevant loss metrics:

- Each of the trained models are then optimized to find their worst-case distributions with respect to our provided decision problems.

- For each worst-case distribution (defined by a unique combination of predictive model, loss function, and optimization instance) we run 15 iterations of our Frank-Wolfe algorithm, with a momentum value of $p_t = 0.7$ and with 35,000 optimization problems sampled per iteration.

- For each of the three worst-case distributions converged for each of the models, we evaluate the expected performance of the distribution on all three loss functions, given the original model used to generate the distribution. This is accomplished by sampling 200 optimization instances from each distribution/loss function combination, sampling 4000 decision problems from each optimization instance, and aggregating over distribution/loss function pairs. For all models we set variables $\rho_1 = n_j = 40, \rho_2 = 6.25$ in order to impose meaningful constraints on optimization problems (i.e., balance between small $\rho$ values that do not allow for significant deviation from the empirical distribution and large values yielding unconstrained optimization over the simplex).

Regarding our experiment comparing our method to Pyomo/IPOPT:

- In this experiment we set $n_j = 8$. Note that the degree of the polynomial scales with $n_j$. As a result traditional polynomial solvers such as Pyomo/IPOPT can become computationally-consuming at large scales. This is particularly true when many calls are made to the solver (in our case, Pyomo/IPOPT is called once for each worst-case distribution w.r.t. each metric).

- Each colored curve in Figure 4 scales up to 3,000 samples per iteration of the Frank-Wolfe algorithm.

# E ADDITIONAL PRESENTATION OF AGGREGATE RESULTS

Below we include, for all combinations of dataset and method used for training the underlying predictive model (CE/MSE, SPO), a set of aggregate results identical to those of Figure 1 but with 95% confidence bounds. Note that for each individual subfigure, the confidence interval is calculated using a normality assumption, with the results of the 50 individual predictive models belonging to that subfigure's combination of dataset and training method serving as data points.

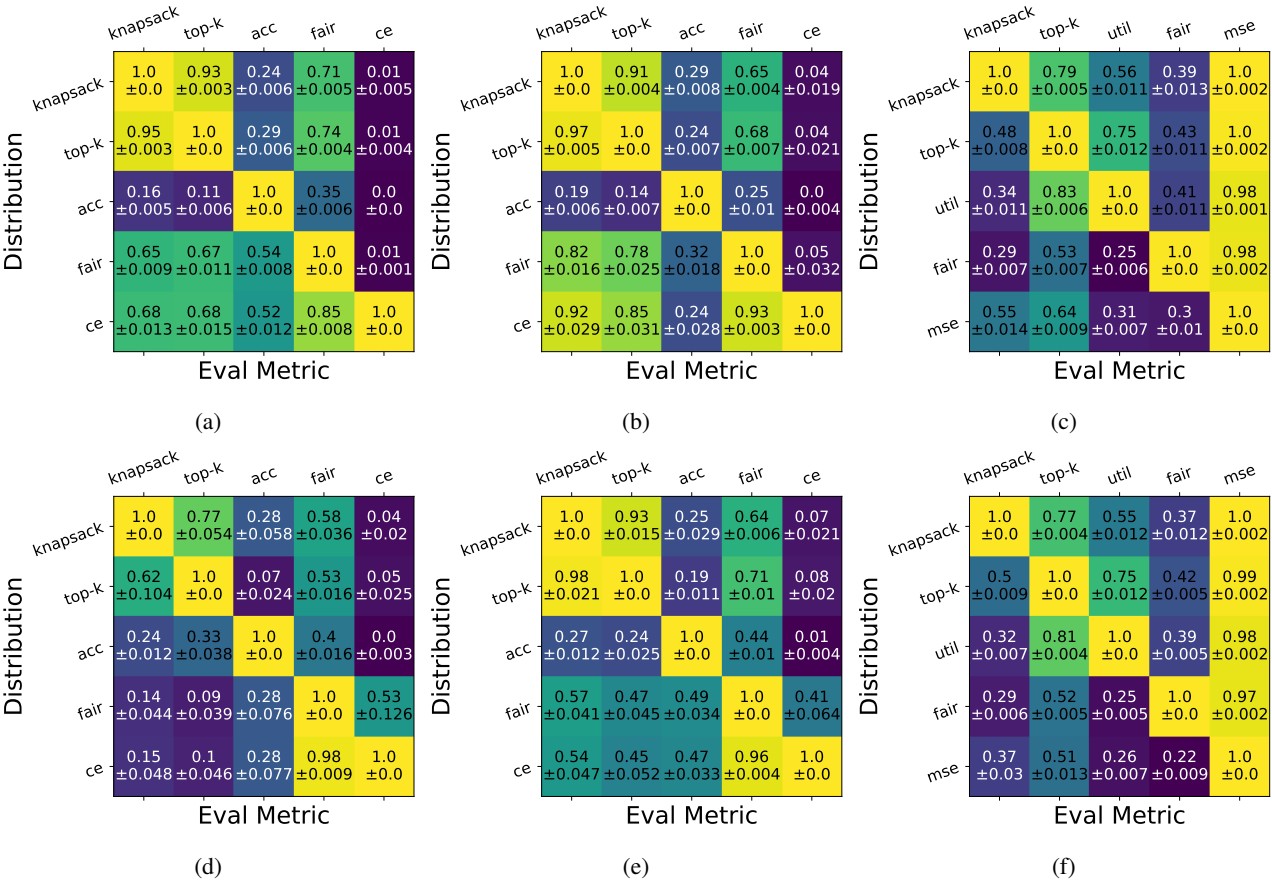

Figure 5: Diagonal-normalized aggregated heat maps with 95% confidence intervals over states for models trained with CE loss (in the regression case, mean-squared error) (top row) and SPO loss (bottom row). From left to right in each row, results are displayed by task for (a,d) employment classification, (b,e income classification, and (c,f) income regression. Within each heat map, rows denote the metric the worst-case distribution maximizes, and columns denote the metrics the worst-case distribution was evaluated on. Note that each column is divided by the diagonal entry in that column, resulting in a main diagonal of all 1.0. Furthermore, because CE loss is always negative, each entry in columns corresponding to CE loss is equal to the diagonal entry in that column divided by the original loss in that cell.

