# OpenReview forum: "Decision-Focused Evaluation of Worst-Case Distribution Shift"
_auai.org/UAI/2024/Conference — UAI 2024 poster_

### Official Review · Reviewer_t4MY · 2024-03-18

**Q2-1 Originality-Novelty:** 3
**Q2-2 Correctness-Technical Quality:** 3
**Q2-5 Clarity Of Writing:** 2

**Q1 Summary And Contributions:**

This paper investigates the distribution shift in resource allocation tasks, offering a fine-grained model of the problem. It introduces a framework designed to capture the distribution shift both within and across instances. Specifically, the framework is supported by a two-level generative model and introduces a novel distribution robust optimization objective. Furthermore, the authors reformulate the proposed objective into a distributionally robust submodular optimization problem, efficiently solvable via the Frank-Wolfe algorithm. The findings are substantiated through comprehensive experiments.

**Q2-3 Extent To Which Claims Are Supported By Evidence:**

3: Good: the main claims are supported by convincing evidence (in the form of adequate experimental evaluation, proofs, (pseudo-)code, references, assumptions).

**Q2-4 Reproducibility:**

2: Fair: key resources (e.g. proofs, code, data) are unavailable but key details (e.g. proof sketches, experimental setup) are sufficiently well-described for an expert to confidently reproduce the main results.

**Q3 Main Strengths:**

This paper is motivated by a good reason to offer a more detailed depiction of the resource allocation problem, especially in the context of potential distribution shifts. The authors systematically formalize their concepts and present a coherent solution. Additionally, experiments are conducted to validate their findings.

**Q4 Main Weakness:**

Enhancing the readability of this work could be achieved if the authors define key terminologies, such as "instance" and "individual," at the beginning of the introduction. Additionally, this paper lacks a comparison of their methods with those of previous studies.

**Q5 Detailed Comments To The Authors:**

- Is the definition of the first equation in page 4 correct?
- Comparing your results with previous works, especially those that predict outcomes without detailed consideration of the instances, could be beneficial.
- It is appreciated if the authors could explain how the algorithm balances the trade-off between resource allocation and prediction accuracy.

**Q9 Complying With Reviewing Instructions:**

Yes

---

> ### Author Rebuttal · Authors · 2024-04-04
>
> Thank you for your thoughtful review! We will try to address your main concerns below:
>
> > “define key terminologies, such as "instance" and "individual" [for better readability]”
>
> Thank you for pointing this out; we have more formally defined these terms in the Introduction!
>
> To more clearly define these terms here, an ‘individual’ in our setting refers to a single row of tabular data that is either treated or not treated in an allocation task. For the Folktables and COVID data this would represent a single person who wishes to receive income/employment assistance, or COVID treatment, respectively. Each individual belongs to a single optimization instance, which are segmentations of all the individuals into subpopulations (e.g. people who live in the same state, or people who came into the hospital in the same month). These instances represent the sets of people who will be considered for treatment in the same optimization problem (e.g. be in the same input to top-k or knapsack).
>
> > “Lacks a comparison [with previous methods]”
>
> Our work is the first to account for downstream optimization in assessing worst-case distribution shift. The closest comparison to previous work is those that define the worst-case distribution with respect to an individual-level loss [1,2]. We compare our results to these past works via 1-accuracy loss in the experiments, which can be seen as implementing the standard f-divergence approach popularized by Namkoong and Duchi [3]. As noted in the Discussion, we find that worst-case accuracy is a poor proxy for worst-case loss on downstream optimization tasks. To further address your concern, we have updated our draft to emphasize the effectiveness of our method, in comparison to prior methods.
>
> > “[correctness of] definition of first equation in page 4”
>
> We believe that the definition of Equation 1 is correct (and the other equations on Page 4), but are happy to answer any questions concerning their derivation or motivation!
>
> > “ trade-off between resource allocation and prediction accuracy”
>
> In our method the user specifies their desired loss, which could include a specified tradeoff between allocation quality and prediction accuracy. For instance, the user could define the loss to be decision loss + $\lambda$*prediction loss for a chosen value of $\lambda$, to put more emphasis on one or the other.
>
> We hope these answers help clarify your concerns. We would be happy to answer any further questions you have. If you do not have any further questions, we hope that you may consider raising your score. Thank you again for your constructive feedback!
>
> [1] Subbaswamy, Adarsh, Roy Adams, and Suchi Saria. "Evaluating model robustness and stability to dataset shift." International conference on artificial intelligence and statistics. PMLR, 2021.
>
> [2] Li, Mike, Hongseok Namkoong, and Shangzhou Xia. "Evaluating model performance under worst-case subpopulations." Advances in Neural Information Processing Systems 34 (2021): 17325-17334.
>
> [3] Namkoong, Hongseok, and John C. Duchi. "Stochastic gradient methods for distributionally robust optimization with f-divergences." Advances in neural information processing systems 29 (2016).

---

### Official Review · Reviewer_jEYx · 2024-03-22

**Q2-1 Originality-Novelty:** 3
**Q2-2 Correctness-Technical Quality:** 2
**Q2-5 Clarity Of Writing:** 3

**Q1 Summary And Contributions:**

This paper studies the task of predictive resource allocation (PRA), in a novel and general setting, where the distribution shifts at the instance level rather than the individual level. The authors proposed a model for their setting, wherein the individual distribution is determined by a latent parameter shared within each instance. Based on their model, the authors propose an efficient variation of Frank-Wolfe algorithm under a non-monotone submodular problem. The authors also conduct extensive experiments, validating the significance of the setting they raised, shedding insights on the coupling of machine learning models and downstream decision tasks, especially for the predict-then-optimize scheme and real-world scenarios.

**Q2-3 Extent To Which Claims Are Supported By Evidence:**

2: Fair: the main claims are somewhat supported by evidence (but the experimental evaluation may be weak, or does not match entirely with the claims, important baselines may be missing, proofs contain important ideas but lack rigor, algorithmic details are only discussed superficially, references are imprecise, assumptions are not sufficiently motivated or explicated, etc.).

**Q2-4 Reproducibility:**

2: Fair: key resources (e.g. proofs, code, data) are unavailable but key details (e.g. proof sketches, experimental setup) are sufficiently well-described for an expert to confidently reproduce the main results.

**Q3 Main Strengths:**

Strengths

[S1] This paper points out and empirically verifies that distribution shifts at an instance level and an individual level are fundamentally different for PRA tasks and have a significant impact on the performance of upstream model. The relation between downstream task evaluation metric on the upstream model training metric is insightful, especially when the upstream model is already designed to be robust to distribution shifts.

[S2] The writing is clear and easy to read. The paper provides rigorous formalization and adequate analysis on experiments results.

**Q4 Main Weakness:**

Weaknesses

[W1] The authors claim that the main contribution of the paper is to propose an end-to-end pipeline, but do not discuss the effectiveness of their methods (for example, compared with allocations methods that are employed in network allocation, etc.).

[W2] The authors model the individual distribution shift as a probabilistic graphical model containing latent variables and conducted experiments, but do not verify that the real-world data conformed to this assumption.

**Q5 Detailed Comments To The Authors:**

My questions mainly concern the experiment, as stated in the Weaknesses section.

[Q1] What is the author's main motivation? If the main motivation is to propose a pipeline for use, is the proposed pipeline better than previous allocation methods in identifying worst-case shifts? If the main motivation is to point out the insufficient emphasis on instance-level shifts in PRA, is the pipeline proposed in this article representative?

[Q2] The two-level generative model is a relatively ideal assumption, and the time series formed by $\xi_j$ is not likely to be i.i.d.. Can the experiment results be attributed to the two-level generative model not adequately modeling real-world shifts?

**Q9 Complying With Reviewing Instructions:**

Yes

---

> ### Author Rebuttal · Authors · 2024-04-04
>
> Thank you for your thoughtful review! We will try to address your main concerns below:
>
> > “main contribution is … end-to-end pipeline, but do not discuss the effectiveness of their methods (for example, compared with allocations methods…)”
>
> Our work is the first to account for downstream optimization in assessing worst-case distribution shift. The closest comparison to previous work is those that define the worst-case distribution with respect to an individual-level loss [1,2]. We compare our results to these past works via 1-accuracy loss in the experiments, which can be seen as implementing the standard f-divergence approach popularized by Namkoong and Duchi [3]. As noted in the Discussion, we find that worst-case accuracy is a poor proxy for worst-case loss on downstream optimization tasks. To further address your concern, we have updated our draft to emphasize the effectiveness of our method, in comparison to prior methods.
>
> > “What is the main motivation?...”
>
> Thank you for raising this clarification point! Our motivation is best described by your first statement, where we propose a pipeline for identification of worst-case losses wrt decision-based metrics. We compare our pipeline to previous methods by including individual-level metrics (1-accuracy) in our empirical evaluation. Note that previous works [1,2] focus only on worst-case shift wrt such standard loss functions, and we find that these are not a good proxy for common decision tasks. Thus, the empirical conclusion from our experiments is that, because previous work only considers these standard loss functions, they are insufficient to handle instance-level worst-case shifts for decision-based tasks. We have updated our Introduction to articulate this motivation more clearly.
>
> > “[models] individual distribution shift as a PGM containing latent variables… but do not verify that the real-world data conformed to this assumption.”
>
> One main underlying assumption is that after conditioning on $\xi$, individuals are drawn iid (e.g. individuals in the same area or month of treatment are iid). Unfortunately, it is not possible to test whether data is iid without additional assumptions, so determining this requires domain knowledge. However, iid assumptions are made near-universally in ML and we have no reason to believe that they fail specifically in our applications (e.g. there is no reason to think that census responses from different households are correlated). In the response to your question below, we elaborate on how if this assumption were to fail, it is unlikely to change our main conclusions.
>
> > “The two-level generative model is a relatively ideal assumption … Can the experiment results be attributed to the model not adequately modeling real-world shifts?”
>
> First, we remark that even if the generative model is incorrect, our results *still* provide a lower bound on worst-case loss. This is because our method optimizes over product distributions. If the assumption that individuals are conditionally iid is invalid, the true uncertainty set includes both product distributions and distributions with correlated draws. Lower bounds are useful in some settings, e.g. to demonstrate that a model is not robust.
>
> Moreover, we expect that our empirical conclusions would only be strengthened if the conditional iid assumption fails. The worst-case distribution wrt decision-blind metrics does not change if we allow correlations (formally, the loss is an expectation over individuals, and linearity of expectation implies that introducing correlations does not change expected loss). By contrast, optimization problems may be sensitive to correlations (see e.g. [4]) that could now be included in the worst-case distribution. Accordingly, the *true* gap between the worst-case decision-blind and worst-case decision-based losses would only increase given an incorrect iid assumption, *relative to the results we presented*. Therefore our results, if anything, **understate** rather than overstate the distinction between worst-case losses wrt decision-blind and decision-based metrics. This reinforces our conclusion that evaluating robustness for specific downstream decision tasks is necessary in predictive resource allocation.
>
> We hope these answers help clarify your concerns. We would be happy to answer any further questions you have. If you do not have any further questions, we hope that you may consider raising your score. Thank you again for your constructive feedback!
>
> [1] Subbaswamy, et al. "Evaluating model robustness and stability to dataset shift." AISTATS 2021.
>
> [2] Li, et al. "Evaluating model performance under worst-case subpopulations." NeurIPS 2021.
>
> [3] Namkoong, et al. "Stochastic gradient methods for distributionally robust optimization with f-divergences." NeurIPS 2016.
>
> [4] Cameron, Chris, et al. "The perils of learning before optimizing." AAAI 2022.

---

### Official Review · Reviewer_4LnA · 2024-03-27

**Q2-1 Originality-Novelty:** 3
**Q2-2 Correctness-Technical Quality:** 3
**Q2-5 Clarity Of Writing:** 3

**Q1 Summary And Contributions:**

This work proposes to identify the worst-case distribution in resource allocation tasks. Although such idea has been raised in prediction tasks, this paper adopts it to the decision-making and gets interesting observations. The experimental results make sense, further validating the soundness of this paper.

**Q2-3 Extent To Which Claims Are Supported By Evidence:**

3: Good: the main claims are supported by convincing evidence (in the form of adequate experimental evaluation, proofs, (pseudo-)code, references, assumptions).

**Q2-4 Reproducibility:**

2: Fair: key resources (e.g. proofs, code, data) are unavailable but key details (e.g. proof sketches, experimental setup) are sufficiently well-described for an expert to confidently reproduce the main results.

**Q3 Main Strengths:**

- The idea of investigating the worst-case distribution for decision-making problems is interesting and novel.
- The experimental results make sense and provide insights for studying the distribution shift problem in the context of resource allocation.

**Q4 Main Weakness:**

The findings make sense, since for different resource allocation tasks, it is similar to different loss functions. Therefore, the worst-case distributions are expected to differ. The authors design / simulate resource allocation tasks based on real-world prediction tasks. I would like to know whether there are some open-sourced real-world datasets for decision-making problems. And I hope to see some results on real decision-making tasks, which may  better support the claims.

**Q5 Detailed Comments To The Authors:**

Please refer to the weaknesses.

**Q9 Complying With Reviewing Instructions:**

Yes

---

> ### Author Rebuttal · Authors · 2024-04-04
>
> Thank you for your thoughtful review! We will try to address your concerns below:
>
> > “[are] there are some open-sourced real-world datasets for decision-making problems? [want] to see some results on real decision-making tasks”
>
> We find your point regarding finding datasets using real-world allocation tasks to be intriguing, and certainly something that would strengthen our findings. Unfortunately, to our knowledge no such publicly available dataset exists that allows us to test our method on real decision-making tasks. This is because such a dataset would not allow us to view counterfactual outcomes (e.g. what would have happened if an individual had received the opposite treatment status), which would be a clear consequence of using a dataset consisting of real-world decision-making outcomes. In this setting, having access to these counterfactuals would be necessary to validate the effectiveness of our method, which is **not a concern with our observational data** since no treatments have been induced. However, we would like to emphasize that our COVID data is closely related to such a real-world task – the problem of developing equitable methods of Paxlovid allocation is a topical example of one real-world allocation problem with a rich body of work surrounding it, in the realm of COVID research [1,2,3].
>
> We hope these answers help clarify your concerns. We would be happy to answer any further questions you have. If you do not have any further questions, we hope that you may consider raising your score. Thank you again for your constructive feedback!
>
> [1] Sullivan, Meg. "Notes from the field: dispensing of oral antiviral drugs for treatment of COVID-19 by ZIP code–Level Social Vulnerability—United States, December 23, 2021–August 28, 2022." MMWR. Morbidity and Mortality Weekly Report 71 (2022).
>
> [2] Tarabichi, Yasir, David C. Kaelber, and J. Daryl Thornton. "Early racial and ethnic disparities in the prescription of nirmatrelvir for COVID-19." Journal of General Internal Medicine 38.5 (2023): 1329-1330.
>
> [3] Haendel, Melissa A., et al. "The National COVID Cohort Collaborative (N3C): rationale, design, infrastructure, and deployment." Journal of the American Medical Informatics Association 28.3 (2021): 427-443.

---

### Official Review · Reviewer_WTTC · 2024-03-29

**Q2-1 Originality-Novelty:** 2
**Q2-2 Correctness-Technical Quality:** 3
**Q2-5 Clarity Of Writing:** 3

**Q1 Summary And Contributions:**

The article introduces a new framework to study optimal resource allocation in the presence of distributional shifts.

**Without distributional shifts**, the decision maker would face the following problem.
1. Learn a predictive model $\hat Y:= m(X)$ for $Y$;
2. Allocate a constrained vector of resources $Z$ by solving $Z^*(\hat Y) := \arg\max_Z f(Z, \hat Y)$, where $f$ is an objective function;
3. Evaluate the *decision loss* with $DL(Y, \hat Y) := g(f(Z^*(Y),  Y), f(Z^*(\hat Y), Y))$, where $g$ is a regret function;
4. Evaluate the expected decision loss as $E_{(X, Y) \sim P} DL(Y, m(X))$.

**With distributional shift**, the decision maker replaces step 4. by minimizing the worst-case expected decision loss (WEDL) $\max_{P \in \Theta} E_{(X, Y) \sim P} DL(Y, m(X))$, where $\Theta$ is an uncertainty set of distributions.

**Generative model**:
The authors model the problem as a two-step generative model. In particular, the individuals are split into $k$ (not necessarily independent) groups. For each of the $k$ groups: i) a latent variable is sampled as $\xi \sim P_\xi$; ii) the predictors and target variable are sampled as $(X, Y) \sim P_{X, Y \mid \xi}$.

**Uncertainty set**: Given the generative model, they characterize the uncertainty set $\Theta$ by allowing shifts both at the latent variable level, $P_\xi$, and at the predictor-response level $P_{X, Y \mid \xi}$.

The authors show the WEDL is non-convex and reformulate it as a submodular optimization problem.

The article concludes with experiments showing that WEDL can differ significantly from classical worst-case expected prediction loss.

**Main contribution**:
This article's main contribution is to study the effect of distributional shifts on the downstream task of allocating contained resources (i.e., worst-case expected decision loss) rather than focusing on worst-case expected prediction loss, which is the target of inference in classical out-of-distribution generalization articles.

**Q2-3 Extent To Which Claims Are Supported By Evidence:**

3: Good: the main claims are supported by convincing evidence (in the form of adequate experimental evaluation, proofs, (pseudo-)code, references, assumptions).

**Q2-4 Reproducibility:**

3: Good: key resources (e.g. proofs, code, data) are available and key details (e.g. proofs, experimental setup) are sufficiently well-described for competent researchers to confidently reproduce the main results.

**Q3 Main Strengths:**

The article's main strength lies in its novel framework for studying distributional shifts in resource allocation problems and comparing the effect of distribution shifts on decision loss rather than predictive loss.

**Q4 Main Weakness:**

The main weakness lies in the characterization of the set of shifts.
In particular, it is unclear why the worst-case expected decision loss can be expressed as a sum (and not an integral) in the problem. For example, by using the Wasserstein distance, the uncertainty set around the empirical distribution can, in principle, contain continuous distributions with respect to the Lebesgue measure.

**Q5 Detailed Comments To The Authors:**

Related to Q4, could it be possible to clarify why the worst-case decision loss can be written as a sum? Is this perhaps due to the specific choice of the divergence?

**Q9 Complying With Reviewing Instructions:**

Yes

---

> ### Author Rebuttal · Authors · 2024-04-04
>
> Thank you for your thoughtful review! We will try to address your concerns below:
>
> > “main weakness in…characterization of the set of shifts…why the worst-case expected decision loss can be expressed as a sum?”
>
> This is a generic property of f-divergences such as the chi-squared divergence (i.e., not specific to our method) because their support only contains points observed in the original sample. This means that the distribution can be parameterized as a probability mass function that reweights the observed samples. As such, we express worst-case expected decision loss as a weighted sum of losses over samples of size $n_j$ from a given optimization instance. Extending our framework to Wasserstein divergence is an interesting future question; these are more expressive but typically more computationally costly. Accordingly, f-divergences have been very commonly used in machine learning applications of distributionally robust optimization (e.g. 1-3).
>
> [1] Namkoong, Hongseok, and John C. Duchi. "Stochastic gradient methods for distributionally robust optimization with f-divergences." Advances in neural information processing systems 29 (2016).
>
> [2] Hashimoto, Tatsunori, et al. "Fairness without demographics in repeated loss minimization." International Conference on Machine Learning. PMLR, 2018.
>
> [3] Staib, Matthew, Bryan Wilder, and Stefanie Jegelka. "Distributionally robust submodular maximization." The 22nd International Conference on Artificial Intelligence and Statistics. PMLR, 2019.

---

### Official Review · Reviewer_xJRV · 2024-03-29

**Q2-1 Originality-Novelty:** 2
**Q2-2 Correctness-Technical Quality:** 3
**Q2-5 Clarity Of Writing:** 2

**Q1 Summary And Contributions:**

The paper proposes an algorithm to find the worst-case distribution shift for predictive resource allocation tasks. The paper argues the use of a downstream decision loss for identifying worst-case distribution shifts instead of basing it on the model accuracy.

**Q2-3 Extent To Which Claims Are Supported By Evidence:**

3: Good: the main claims are supported by convincing evidence (in the form of adequate experimental evaluation, proofs, (pseudo-)code, references, assumptions).

**Q2-4 Reproducibility:**

3: Good: key resources (e.g. proofs, code, data) are available and key details (e.g. proofs, experimental setup) are sufficiently well-described for competent researchers to confidently reproduce the main results.

**Q3 Main Strengths:**

The paper tackles an important problem in machine learning: robust deployment of ML systems. A key step for achieving robust models is identifying worst-case distribution shifts from some ambiguity set, which is used, for instance, in distributionally-robust methods. This paper makes a valid and important point that we need to take the downstream task into account while doing so. I totally agree with that sentiment. The paper seems fairly sound overall, although I did not check the proofs in detail. The authors empirically show that worst-case distributions are different when different loss functions are used, which was the main claim of the paper.

**Q4 Main Weakness:**

I felt that the paper's clarity can be improved, as it prevented me from understanding the method and the experimental observations in detail, which was a bit of a let down considering that the introduction was pretty nicely written and explained. At times in Section 2 and Section 3 (especially) I felt that the authors could have been more rigorous and write the expressions/equations in addition to saying in words. For instance, I found the last para before Section 4 too dense to understand. Either this is an important part of the technical novelty, which should then be explained in more details with equations, or some computational/implementation trick which the reader doesn't need to care about and can be moved to the appendix. Currently it is part of the method section, but is too technical with many new terms to make any sense of it (my unfamiliarity with the subject matter is partially contributing to it I am sure).

A tip: maybe use the simple example from the introduction as a running example throughout the paper to concretize the different notation/quantities to a real-world setting.

The worst-case distribution plots were a bit difficult for me to interpret, and the explanations did not help much. The only thing I managed to get was that they are different for the different losses. The empirical results lack some sort of uncertainty quantification across different runs. Doesn't it make sense to have some standard deviation for the aggregated results? How much do the worst-case distribution plots vary if you re-run the method?

The choice of the three loss functions seem a bit arbitrary. What is it based on? If they are common losses used in the literature, please cite a few papers that used them.

It is not clear if the proposed algorithm has any hyperparameters (no need to select \rho_\xi?),  and there is no discussion on how to select them or experiments checking the sensitivity of the results to their choices. (I would assume that if the method had no hyperparameters, that would be quite a unique selling point and would have been mentioned).

Please mention any limitations of the proposed work.

**Q5 Detailed Comments To The Authors:**

- What do the authors mean by "end-to-end pipeline". It was never mentioned beyond the introduction.
- The authors claim that they are the first to "examine the consequences of this divergence for distribution shift", but in the additional related work paragraph, they mention Mo et al.[2021], which seems quite related as per them ("differs in considering the consequences of joint optimization..."). Is it still fair to claim that you are the first?
- "We remark that some appropriate \xi is guaranteed to exist by De Finetti's theorem...". How? I would expect a reference here at least if there is no explanation (or is it common knowledge?).
- Does your algorithm require to set \rho_\xi to some value? If yes, how can one go about choosing an appropriate value? How did you set it for the experiments?(I did not go through the appendix in detail so I may have missed this).
- "Our solution requires that our feasible set contains the zero vector". Why? Can you give some intuition?
- What do you mean by "equivalent *quality* approximation"?
- How sensitive is the method to varying hyperparameters (if any)?
- In Figure 1a, why is the accuracy-accuracy result is so low (0.74)?
- "...in Figure 3d, we observe higher converged probabilities...between 0.8 and 0.9 than in Figure 3c". Frankly, I do not see any difference visually. What am I missing?
- Figures 4b and 4d look very similar even though one is accuracy loss and the other is decision-based. Why?
- In the COVID-19 experiment, 1-accuracy led to discrete segmentations of the data while there was continuous relationship for decision-based losses. What's the implication of this observation? Why is the continuous one better than piecewise one? (I was quite lost here, but hopefully my question makes some sense).
- The implications of the work are not outlined in the conclusion/discussion. What do the authors think is the next step? Combining the proposed method with some distributionally robust frameworks?

Minor comment:
- Paragraph 2 of introduction: Duchi and Namkoong [2021] should be \citep I think.

**Q9 Complying With Reviewing Instructions:**

Yes

---

> ### Author Rebuttal · Authors · 2024-04-04
>
> Thank you for your thoughtful review! We will try to address your concerns below:
>
> > Clarity of writing
>
> Thank you for your suggestions! We have updated our draft with the suggested revisions.
>
> > Uncertainty quantification
>
> We compute 95% [confidence intervals](https://imgur.com/a/OyWSBc5) for Folktables data, using results for each trained model (e.g. Figure 3a) as data points. Almost all pairwise comparisons between intervals are statistically significant and widths are generally within 0-2%. As such our conclusions are robust to sampling noise. We have updated the draft to include these results. Note for COVID data, only 1 SPO/CE model was trained, so confidence bounds were not motivated.
>
> >  Choice of loss functions
>
> First, classification accuracy is among the most common assessments of predictive performance. Knapsack and top-k are standard models of constrained allocation. Specifically, knapsack is widely used in the predict-then-optimize literature [1,2,3]). Top-k is a simplification of knapsack where cost is identical for all individuals. This helps us understand how knapsack’s cost function influences the underlying decision problem.
>
> > Hyperparameter Selection
>
> $\rho_\xi$ and $\rho_\text{ind}$ are user-chosen parameters. However rather than tuning them to maximize performance, users select $\rho$ to describe how much distribution shift to allow. It is common in worst-case analysis under distribution shift to have a similar parameter based on anticipated shift [4,5,6].
> We use $\rho_\xi=6.25, \rho_\text{ind}$=500, which correspond to an intermediate or practical level of shift (at low levels, the distribution is unable to change; at high levels the distribution can be arbitrary). We conducted [additional experiments](https://imgur.com/a/9pmE3Kd) to see how results vary with $\rho$. Our high-level results are independent of $\rho$; each grid contains large main diagonal entries with weak off-diagonals, meaning worst-case optimization wrt individual-level metrics (acc) fails to approximate worst-case outcomes for decision-based tasks (top-k). Variations in $\rho$ yield intuitive results; increasing $\rho_\xi$ or $\rho_\text{ind}$ results in higher losses, as the optimization problem is less constrained.
>
> > Limitations
>
> Our framework requires domain knowledge to specify the size of the uncertainty set using $\rho$. While this is common to most models of distribution shift [4,5,6], we will elaborate on it in the conclusion, including how existing methods for providing interpretable sensitivity parameters in DRO might be adapted for predictive resource allocation [7].
>
> > “[claim that] first to "examine the consequences of this divergence for distribution shift"...Mo et al.[2021] seem related”
>
> The key distinction is that our work is the first to consider the consequences of *joint optimization* over a group of individuals (where constraints imply that one individual’s prediction may impact another individual’s decision). Mo et al [2021] consider individual-level treatment assignments, in line with previous work [8,9]. We will clarify this in our draft.
>
> > “ $\xi$ is guaranteed to exist by De Finetti's theorem”
>
> Thanks for raising this point; we have added the following citation [10]. De Finetti’s theorem is commonly invoked this way, e.g. in the Bayesian nonparametrics literature.
>
> > Figure 1a low accuracy-accuracy
>
> The in-distribution misclassification rate (1-accuracy) of our predictive models is roughly 20%. Since 1a shows this error increases to 74%, our results reveal a significant performance drop.
>
> > “Figure 3d [has] higher probabilities ... than Figure 3c”
>
> 3d has larger weights (lighter-colored) on positives in the band between y=.8 and y=1.0 than 3c. This means the worst-case distribution wrt knapsack samples more high-prediction (likely to be treated) positives, lowering loss on decision-based tasks.
>
> > [COVID-19] Why is continuous better than piecewise?
>
> In 5a (noncontinuous), false negatives with low predictions (e.g. 0.55) (type A) are as likely to be sampled as negatives with high predictions (e.g. 0.99) (type B). In 5b (continuous), type B, who will more often be treated in downstream allocation, is sampled more than type A. This means distributions like 5b are better-suited for achieving high loss on decision-based tasks.
>
> [1] Mulamba, et al. Contrastive losses and solution caching for predict-and-optimize.
> [2] Stuckey, et al. Dynamic programming for predict+optimise.
> [3] Tang, et al. Pyepo.
> [4] Namkoong, et al. Stochastic gradient methods for DRO with f-divergences.
> [5] Lam. Robust sensitivity analysis for stochastic systems.
> [6] Staib, et al. Distributionally robust submodular maximization.
> [7] Namkoong, et al. Minimax Optimal Estimation of Stability Under Distribution Shift.
> [8] Subbaswamy, et al. Evaluating model robustness and stability to dataset shift.
> [9] Li, et al. Evaluating model performance under worst-case subpopulations.
> [10] Orbanz, et al. Bayesian Nonparametric Models.

---

### Meta-Review · Area_Chair_ByfF · 2024-04-16

The five reviews for this paper are leaning to support its acceptance, but not strongly (all scores are borderline-weak). Although the reviews identified some strengths of the work, it appears that there weaknesses that could affect the ability of readers to understand the work and to appreciate its contributions. The said weaknesses are significant enough to raise doubts about the improvements that could be expected in a single round of reviews, suggesting that this paper could benefit from an additional round of reviews before it is ready for publication. I think this explains the borderline to weak scores.